# Dynamic mRNP Remodeling in Response to Internal and External Stimuli

**DOI:** 10.3390/biom10091310

**Published:** 2020-09-11

**Authors:** Kathi Zarnack, Sureshkumar Balasubramanian, Michael P. Gantier, Vladislav Kunetsky, Michael Kracht, M. Lienhard Schmitz, Katja Sträßer

**Affiliations:** 1Buchmann Institute for Molecular Life Sciences, Goethe University Frankfurt, 60438 Frankfurt a.M., Germany; kathi.zarnack@bmls.de; 2School of Biological Sciences, Monash University, Clayton, VIC 3800, Australia; sureshkumar.balasubramanian@monash.edu; 3Centre for Innate Immunity and Infectious Diseases, Hudson Institute of Medical Research, Clayton, VIC 3168, Australia; michael.gantier@hudson.org.au; 4Department of Molecular and Translational Science, Monash University, Clayton, VIC 3800, Australia; 5Institute of Biochemistry, FB08, Justus Liebig University, 35392 Giessen, Germany; vladislav.kunetki@chemie.bio.uni-giessen.de; 6Rudolf Buchheim Institute of Pharmacology, FB11, Justus Liebig University, 35392 Giessen, Germany; Michael.Kracht@pharma.med.uni-giessen.de; 7Institute of Biochemistry, FB11, Justus Liebig University, 35392 Giessen, Germany; Lienhard.Schmitz@biochemie.med.uni-giessen.de

**Keywords:** signal transduction, extracellular stimuli, intracellular stimuli, stress, post-transcriptional gene regulation, RNA-binding proteins (RBPs), messenger ribonucleoprotein particles (mRNPs)

## Abstract

Signal transduction and the regulation of gene expression are fundamental processes in every cell. RNA-binding proteins (RBPs) play a key role in the post-transcriptional modulation of gene expression in response to both internal and external stimuli. However, how signaling pathways regulate the assembly of RBPs with mRNAs remains largely unknown. Here, we summarize observations showing that the formation and composition of messenger ribonucleoprotein particles (mRNPs) is dynamically remodeled in space and time by specific signaling cascades and the resulting post-translational modifications. The integration of signaling events with gene expression is key to the rapid adaptation of cells to environmental changes and stress. Only a combined approach analyzing the signal transduction pathways and the changes in post-transcriptional gene expression they cause will unravel the mechanisms coordinating these important cellular processes.

## 1. Introduction

Gene expression is a multi-step process involving the synthesis of the messenger (m)RNA during transcription followed by mRNA processing, export to the cytoplasm, protein synthesis and mRNA decay [1]. During processing, the mRNA is capped at its 5′ end, spliced as well as cleaved and polyadenylated at its 3′ end. These and all subsequent events of the mRNA life cycle are highly interconnected and coordinated by multiple feed-back and feed-forward mechanisms to ensure appropriate gene expression and thus protein levels across diverse conditions [2,3,4]. Broadly, gene expression can be considered as basal (constitutive) or inducible. The latter occurs in response to intrinsic rhythms, such as the cell cycle or the circadian clock, or as a reaction to external cues that include changes in metabolic conditions, inflammation or infection, among many others (Figure 1) [5]. Most external stimuli are sensed by specific receptors that are present at exposed surfaces such as the plasma membrane or endosomes to enable the direct contact to extracellular inducers. Alternatively, intracellular infections by viruses are recognized by specific cytosolic receptors such as STING (stimulator of interferon genes) or RIG-I (retinoic-acid inducible gene I), which recognize viral DNA or RNA, respectively [6,7]. The activation of receptors triggers the induction of a particular signaling pathway, which modulates the de novo expression of mRNAs and subsequent steps of post-transcriptional gene regulation.

### 1.1. Nuclear mRNA Processing

mRNAs undergo a highly coordinated life cycle, starting with their synthesis by RNA polymerase II in the nucleus and ultimately ending in different types of specific or unspecific decay. All steps in RNA processing are dynamically regulated, referred to as “post-transcriptional gene regulation”. The integration of various signals at the RNA level allows cells to react to internal and external stimuli and plays and essential role in their adaptation to the environment. Central to all RNA processing steps and their regulation are *trans*-acting RNA-binding proteins (RBPs) that recognize *cis*-regulatory elements in (pre-)mRNAs. Specificity can be achieved by RNA sequence motifs and/or structural folds [8,9]. During RNA processing, numerous RBPs bind to the mRNAs and assemble into mRNA ribonucleoprotein particles (mRNPs) (see Section 1.2). Many signaling pathways modulate mRNP composition and function—often by post-translational modifications (PTMs) of the RBPs such as phosphorylation and ubiquitination (see Section 2.1.1).

Multiple co- and post-transcriptional processing steps occur to obtain a mature mRNA (Figure 2). As soon as the first ~25 nucleotides (nt) of the nascent RNA are transcribed, an N7-methylated guanosine (m^7^G) is added to the first nucleotide via an unusual 5′-5′ triphosphate linkage [10]. The resulting **cap structure** is evolutionarily conserved and protects the mRNA from 5′-3′ exonuclease cleavage. It plays important roles in essentially all downstream processing steps and is also a key mark of self-RNA, differentiating it from pathogenic RNAs [11]. Many of the regulatory functions are mediated by the cap-binding complex (CBC) that binds to factors of pre-mRNA splicing, cleavage and polyadenylation, transcription termination and nuclear RNA degradation. In higher eukaryotes, CBC and the m^7^G cap are also essential for mRNA export by recruiting nuclear export adapters such as ALYREF, a TREX (TRanscription-Export complex) component (see Section 1.2) [12].

The next processing step to occur is **pre-mRNA splicing**, by which introns are excised and exons are ligated in two consecutive transesterification reactions. Splicing mostly occurs co-transcriptionally, although introns can be selectively detained to control the availability of mature mRNAs [13,14]. At the molecular level, splicing is catalyzed by a multi-subunit ribonucleoprotein machinery called the spliceosome which consists of five uridine-rich small nuclear RNAs (U1, U2 and U4-6 snRNAs) and about 200 proteins [15,16]. At most introns, the 5′ and 3′ splice sites, also known as donor and acceptor sites, are marked by GU and AG dinucleotides, respectively. Additional sequence elements include the polypyrimidine tract (Py-tract), a stretch of uridines and cytidines immediately upstream of the 3’ splice site, and the preceding branch point sequence (BPS) with the consensus motif yUnAy (where y is pyrimidine and n any nucleotide) [16]. In the first steps of spliceosome assembly, the U2 snRNP (small nuclear ribonucleoprotein) is recruited to the 3’ splice site by U2AF (U2 auxiliary factor). Simultaneous binding of the U1 snRNP to the 5′ splice site allows to establish contacts between U1 and U2 snRNPs across the exon (“exon definition”), which are subsequently matured into the pre-spliceosomal A complex by contacts across the intron (“intron definition”) [17]. In this complex, the U2 snRNP replaces SF1 (splicing factor 1) at the BPS. Entry of the tri-snRNP U4, U5 and U6 and a series of remodeling steps lead to the active B complex, which is competent for the first step of the splicing reaction, i.e., opening the 3’ splice site and connecting the free intron end to the BPS. Further conformational changes result in the C complex for the second step, namely the ligation of both exons and the release of the intron as a lariat. The post-spliceosomal complex is then disassembled and recycled.

Selective splice site choices in **alternative splicing (AS)** allow for the generation of multiple distinct transcript isoforms and possibly protein products from the same gene. Additionally, intron retention events, in which one or more introns are not removed during splicing, often disrupt the open reading frame and decrease RNA stability. For instance, external signals such as nutrient starvation result in widespread intron retention in ribosomal protein gene pre-mRNAs, which promotes stress resistance and adaptation of the cell [18,19]. AS is regulated by a large set of RBPs that recognize *cis*-regulatory elements in pre-mRNAs and guide the activity of the spliceosome [20]. For instance, members of the family of serine/arginine (SR) proteins generally promote constitutive and alternative splicing (see Section 2.2.2), whereas heterogeneous nuclear ribonucleoproteins (hnRNPs) are often found as splicing-inhibitory factors [21,22]. The splicing-regulatory function of RBPs is strongly shaped by their dynamic assembly into mRNPs and their interactions with each other, including direct competition and cooperative recruitment [23,24]. Importantly, these interactions are modulated by PTMs. A well-studied example is the phosphorylation of SR proteins by signaling-activated kinases (see Section 1.5), which changes their RNA-binding activity, triggers their sequestration into dedicated organelles such as nuclear speckles and profoundly impacts the resulting splicing pattern [25,26]. Co-transcriptional events and chromatin structure further impact on splicing decisions [27].

In the final maturation step during transcription, endonucleolytic cleavage releases the nascent transcript from the transcribing RNA polymerase II followed by the addition of a non-templated poly(A) tail by PAP (poly(A) polymerase). Most human polyadenylation sites are characterized by a hexamer motif, predominantly AAUAAA, which is precisely positioned 10 to 30 nucleotides upstream of the cleavage site, and further flanked by a U-rich upstream and a GU-rich downstream sequence element (DSE and USE, respectively). **Cleavage and polyadenylation (CPA)** at this site require the assembly of four complexes, including CPSF (cleavage and polyadenylation specificity factor), CstF (cleavage stimulatory factor), CFIm (cleavage factor Im) and CFIIm. The hexamer motif and the USE are recognized by CPSF, which also harbors the endonuclease activity, whereas DSE recognition is provided by CstF. CPSF and CstF are sufficient to catalyze CPA in vitro, while CFIm is most critical for the integration of regulatory signals in vivo [28].

The majority of eukaryotic genes are subject to alternative polyadenylation (APA) at more than one polyadenylation site. When APA occurs within the open reading frame, it can affect the encoded protein. More often, APA changes the length of the 3′ UTR (3′ untranslated region) and hence the regulatory potential of mRNAs, impacting on mRNA stability, nuclear export, subcellular localization and/or translation as well as in some cases the localization of the translated protein product [29]. In addition to the core CPA machinery, many RBPs have been identified as potent APA regulators. Interestingly, many of them were first described in splicing, suggesting intricate links between the different RNA processing steps [30,31].

### 1.2. mRNP Assembly and Dynamics

In addition to protein complexes associating transiently with the mRNA for its processing (see Section 1.1 and Figure 2), nuclear mRNA-binding proteins bind stably to the mRNA forming an mRNP. This **mRNP assembly** occurs already co-transcriptionally, i.e., on the nascent mRNA [32,33,34,35,36]. The correct formation of the mRNP is a prerequisite for nuclear mRNA stability and nuclear mRNA export. Furthermore, the protein composition of the mRNP often determines the cytoplasmic stages in the life of the mRNA such as its translation rate, localization and degradation [4,37,38,39]. Thus, correct mRNP assembly is crucial for correct gene expression. Importantly, changes in the mRNP composition are central for cellular adaptation to changing conditions, e.g., caused by environmental stress or during development (Figure 1).

The first complex that is stably deposited on the nascent mRNA is the **CBC**, which binds immediately when the m^7^G cap has been added to the 5′ end of the mRNA (see Section 1.1). The CBC consists of a small and a large subunit, Cbp20 (cap-binding protein 20 kDa subunit) and Cbp80 (cap-binding protein 80 kDa subunit) in *Saccharomyces cerevisiae* and NCBP1 (nuclear cap-binding protein 1) and NCBP2 in human. The cap in complex with the CBC protects the mRNA from degradation and influences transcription elongation, splicing, 3′ end formation and nuclear mRNA export (reviewed in [40]).

Similarly, during splicing (see Section 1.1 and Figure 2), proteins are stably deposited onto the mRNA in a splicing-dependent manner as part of the mRNP assembly process. A prime example is the **exon junction complex (EJC)**, which is deposited onto the mRNA about 20 to 24 nucleotides upstream of each exon–exon junction during splicing [41,42,43]. It consists of the four core subunits eIF4E3 (DDX48), MAGOH, RBM8A (Y14) and Barentsz (BTZ, also referred to as MLN51 or CASC3) [44,45]. The EJC marks the former splice site and carries this information to the cytoplasm [45]. It is best known for its function in initiating the rapid turnover of mRNAs with premature stop codons via nonsense-mediated mRNA decay (NMD) in the cytoplasm [41,43]. Usually, the ribosome removes the EJCs during translation. Since introns are virtually absent from 3′ UTRs, an EJC remaining on a translated mRNA shows that it contains a premature stop codon, i.e., a stop codon that occurs 5′ of the last exon junction. These mRNAs are likely to encode shortened and potentially harmful proteins and are thus quickly degraded (see [46,47,48] for review and references therein). In contrast to its function in NMD, the EJC has positive effects on gene expression by enhancing splicing, nuclear mRNA export and mRNA translation [41,45]. The different functions of the EJC are executed through the recruitment of different EJC-associated proteins [41]. Underpinning the physiological importance of this complex, mutations in EJC components cause disease in humans [49,50].

Another complex that is most likely recruited to the mRNA during splicing in metazoans is the **TREX complex**. TREX was first discovered in *S. cerevisiae*, where it couples transcription to nuclear mRNA export [51]. It enhances transcription elongation, mediates mRNP assembly and is thus required for nuclear mRNA export [51,52]. TREX and its functions are well conserved in many organisms indicating its importance. It consists of the THO complex, which is composed of five subunits in *S. cerevisiae* and six subunits in humans, the RNA helicase Sub2 (UAP56 in human), the mRNA-export adaptor Yra1 (ALYREF) and either the two SR proteins Gbp2 and Hrb1 in *S. cerevisiae* or additional human-specific subunits [51,52,53]. In yeast, the TREX complex is recruited to the transcription machinery by a direct interaction with the transcription elongation-specific form of RNA polymerase II, the Prp19 complex and Mud2, both of which also function in splicing, as well as the mRNA-binding activity of several TREX components [54,55,56,57]. In contrast, TREX is recruited in a splicing-dependent manner in metazoan cells, most likely by the interaction of ALYREF with the EJC component eIF4A3 [58,59]. This splicing-dependent recruitment of TREX might be an explanation for the early finding that splicing enhances mRNA export [60]. Furthermore, TREX is recruited to the 5′ as well as the 3′ end of the mRNA by the interaction of ALYREF with the CBC and PABPN1 (polyA-binding protein nuclear 1), respectively [12,61,62]. This is consistent with the interaction of the ALYREF homolog Yra1 with Pcf11, a component of the cleavage and polyadenylation complex [63]. Taken together, TREX is recruited to the mRNP by multiple protein–protein interactions as well as its interactions with the mRNA. Importantly, TREX plays an important and conserved role in nuclear mRNP assembly as well as other steps of gene expression.

After cleavage and polyadenylation of the mRNA (see Section 1.1 and Figure 2), the poly(A) tail is covered by the protein **PABPN** in mammals and **Pab1** in *S. cerevisiae*. As Pab1 localizes predominantly to the cytoplasm, the main nuclear poly(A)-binding protein in *S. cerevisiae* is **Nab2**. Nab2 is also evolutionarily conserved and its ortholog is called **ZC3H14** in humans. These proteins not only bind and thus protect the poly(A) tail but also control its length and influence multiple downstream processes [64,65,66,67].

Further proteins that package the mRNA into an mRNP are the conserved protein **Tho1** in yeast and its mammalian homolog **SARNP** (SAP domain-containing ribonucleoprotein; also known as CIP29) as well as members of the serine-arginine-rich (SR) protein family. SARNP and Tho1 interact with the TREX complex and are recruited in a TREX-, splicing- and cap-dependent manner [68,69]. However, the function of SARNP/Tho1 in mRNP packaging and other steps of gene expression still needs to be elucidated. Several **SR proteins** are also components of nuclear mRNPs. In yeast, the SR-like protein **Npl3** is already recruited to the transcription machinery, associates with the mRNA co-transcriptionally and functions in transcription elongation, splicing, 3′ end formation and nuclear mRNA export [55,70,71,72,73]. Similar to Npl3, **Nab2** is a nuclear mRNP component with functions in 3′ end processing, nuclear mRNP assembly and nuclear mRNA export [74,75,76]. Like Npl3 and Nab2, several SR proteins associate with the mRNA in the nucleus of mammalian cells [25]. SR proteins are best known for their function in the regulation of splicing [25] (see Section 1.1), but also regulate 3′ UTR length and are required for nuclear mRNA export [25,77].

Taken together, many proteins constitute nuclear mRNPs. While they were first discovered to play a role in a specific processing event, these proteins usually also function in other nuclear and even cytoplasmic steps of gene expression. Thus, nuclear mRNP assembly is tightly integrated within the gene expression pathway.

### 1.3. Nuclear mRNA Export and Its Regulation

The prerequisites for the export of the mRNA to the cytoplasm are correct mRNA processing and mRNP assembly (see Section 1.1 and Section 1.2 and Figure 2). Several components of the nuclear mRNP act as so-called **mRNA export adaptors**: they recruit the nuclear mRNA exporter, Mex67-Mtr2 in *S. cerevisiae* and NXF1-NXT1 (also known as Tap-p15) in mammalian cells, to the mRNP. Multiple mRNA export adaptors exist in yeast: the TREX complex components Hpr1, Yra1, Gbp2 and Hrb1, Npl3 as well as Nab2 [78,79,80,81,82]. In mammalian cells, SR proteins such as SRSF3 serve as mRNA export adaptors [31,83]. Most likely, these different adaptor proteins serve to increase the low intrinsic RNA binding activity of the nuclear export receptor Mex67-Mtr2/NXF1-NXT1 and function in a redundant but also synergistic manner. The mRNA exporter Mex67-Mtr2/NXF1-NXT1 directly binds to components of the nuclear pore complex (NPC) and transports the mRNP through the NPC to the cytoplasm [84].

Interestingly, nuclear mRNA export is regulated under changing environmental conditions. In *S. cerevisiae* and mammalian cells, export of bulk mRNA is blocked after exposure to several stress conditions such as heat or osmotic stress [36,85,86,87]. In contrast, the export of transcripts induced under these stress conditions is selectively facilitated [86]. This **selective export** is most likely regulated by the dissociation of export adaptors from bulk mRNA [86]—possibly by their post-translational modification. For example, Nab2 is phosphorylated by the yeast MAP kinase Slt2, which reduces its binding to Mex67, and localizes with Yra1 in nuclear foci [87]. Interestingly, though, whereas Slt2 is required for nuclear mRNA accumulation during heat shock, the phosphorylation of Nab2 by Slt2 is not, indicating that Slt2 has other targets than Nab2 [87]. Thus, the mechanism of regulated dissociation of Nab2 from the mRNA and the Slt2-mediated mRNA export block remains unresolved. Instead of the recruitment by export adaptors, Mex67-Mtr2 is directly recruited to the site of transcription in the case of heat shock mRNAs through interaction with the heat shock transcription factor Hsf1 [86]. However, the details of this mRNA export switch and, most importantly, how it is controlled by signal transduction pathways is not understood.

### 1.4. mRNA Translation and Decay in the Cytoplasm

Once in the cytoplasm, mRNAs are translated by ribosomes to produce the encoded proteins. While certain mRNAs are targeted to specific subcellular locations to promote local translation, others are translated throughout the cytoplasm [88,89].

**Translation** occurs in four stages (reviewed in [90]). Initiation starts with the assembly of the 43S preinitiation complex (PIC) comprising the small 40S ribosomal subunit, the initiator methionine tRNA (Met-tRNA_i_) and GTP-bound eIF2 (eukaryotic initiation factor 2), supported by eIF3 and further initiation factors. PIC recruitment to the m^7^G cap of the mRNA is mediated by interactions with cap-bound eIF4F, a trimeric complex of the cap-binding protein eIF4E, the RNA helicase eIF4A and the scaffold protein eIF4G, and further stimulated by interactions of eIF4F with PABP (poly(A) binding protein). The PIC scans along the 5’ UTR for the start codon, where pairing with Met-tRNA_i_ triggers PIC remodeling and joining of the large 60S ribosomal subunit to form the translation-competent 80S ribosome. In a repeated elongation cycle, an aminoacyl-tRNA base-pairs with the mRNA codon in the A (aminoacyl) site of the ribosome. Upon peptide bond formation with the growing polypeptide chain in the P (peptidyl) site, GTP hydrolysis in eEF2 (eukaryotic elongation factor 2) fuels translocation of the ribosome and release of the deacetylated tRNA via the E (exit) site. Finally, a stop codon in the A site is recognized by eRF1 (eukaryotic release factor 1) and eRF3, which trigger polypeptide release, followed by ribosome recycling.

Based on the apparent communication between the two ends of the transcript during translation, exemplified by the eIF4F/PABP interaction, it has long been assumed that mRNAs adopt a ring-like conformation, also known as the “closed-loop” model [91]. Multiple recent reports, however, introduced a refined model implying that translation elongation decompacts the mRNP [92,93,94,95]. **Decompaction** is likely driven by the unfolding of inherent mRNA secondary structures by the translating ribosomes, among other factors. Conversely, stress conditions that impair translation initiation and translation inhibitory drugs further compact the mRNPs, bringing the 5′ and 3′ ends into close proximity [94,95,96]. With prolonged inhibition, the compacted mRNPs proceed to higher-order assemblies, including stress granules and P-bodies, which serve as reservoirs of mRNA storage and decay (see Section 2.1.2).

Translation can be controlled at multiple levels to allow for a rapid adaptation of protein production to internal and external signals [90,97]. For instance, a number of 4E-BPs (eIF4E binding proteins) compete with the m^7^G cap for eIF4E binding and thereby prevent its recruitment to mRNAs and subsequent PIC formation. 4E-BP activity is commonly regulated through phosphorylation by the mTOR (mammalian target of rapamycin) kinase. Similarly, phosphorylation of eIF2α prevents the exchange of GDP to GTP and thus globally shuts down translation. A targeted regulation of individual mRNAs can be achieved through regulatory RBPs that often bind to sequence motifs in the 3’ UTR and other transcript regions [98]. Moreover, microRNAs can modulate the translation of specific mRNAs through RNA-RNA interactions that involve partially complementary sequence elements in the 3’ UTRs (see Section 2.2.1).

**mRNA degradation** frequently starts with deadenylation by the CCR4-NOT exonuclease complex [99] and subsequent decapping (see Section 2.1.2). The mRNA is then degraded from both ends through XRN1 (5′-to-3′ exoribonuclease 1) and the exosome (3′-to-5′) [100]. The cap and the poly(A) tail thus play dual roles in translation and mRNA degradation, since both deadenylation and decapping are inversely related to translation; for instance, increased elongation rates disfavor deadenylation [99]. Moreover, RNA modifications such as N6-methyladenosine affect RNA stability [101]. The degradation of selected mRNAs can be accelerated through destabilizing RNA sequence elements, such as AU-rich elements which are recognized by the RNA stability regulator hnRNPD (also known as AUF1) and others [102]. Alternatively, NMD targets mRNAs for selective decay through nonproductive splicing events.

### 1.5. Principles of Signal Transduction

Each inducible signaling system is composed of unique components and features. However, some general principles such as the control of signal thresholds, amplification processes, signal crosstalk and regulatory feed-back loops can be deduced from the well-studied pro-inflammatory signaling pathways. Various molecular mechanisms control the threshold of inflammatory signaling in order to avoid random activation of this signaling cascade, which could lead to low constitutive, smoldering inflammation and its associated pathophysiological effects. In this context, an example for the control of signaling thresholds is provided by the pleiotropic cytokine tumor necrosis factor-alpha (TNFα). This factor binds to its cognate **membrane receptors** TNFR1 (TNF receptor 1) and TNFR2 to trigger various signaling cascades. Notably, the TNFα ligand itself is highly regulated, and the post-transcriptional regulation of *TNFα* mRNA and protein through AU-rich elements has been widely studied, both mechanistically and for its contribution to the control of the entire process of systemic inflammation [103,104]. *TNFR* mRNAs can undergo alternative splicing to produce soluble receptors, which can compete with membrane-bound TNFRs for ligand binding, thereby limiting the threshold or “noise” of spontaneous TNFα-induced signaling. This dampening mechanism is important when the levels of extracellular TNFα are low and exemplifies how regulated RNA processing can have a profound impact on signaling responses in the cell. In contrast, high levels of TNFα, such as those occurring during acute infection, elicit a strong and powerful cellular response by the creation of larger interaction surfaces, starting with the inducible trimerization of TNFRs after ligand binding [105].

The early signaling events after TNFR activation also illustrate the central involvement of **post-translational modifications (PTMs)**, namely regulatory ubiquitination and phosphorylation. These modifications allow the formation of transient protein/protein interactions, changes of protein conformation or intracellular localization and the direct activation of downstream kinases by phosphorylation of their ATP-binding activation loops [106]. At the intracellular side, trimerization of TNFR1 creates a docking surface allowing the inducible binding of RIPK1 (receptor-interacting protein kinase 1) and of the adaptor protein TRADD (TNFR1-asscoiated DEATH domain protein) [107,108]. The TRADD proteins serve as a platform for the recruitment of TRAF2 (TNFR-associated factor 2) and/or TRAF5 proteins that in turn allow binding of the E3 ubiquitin ligases cIAP1 (inhibitor of apoptosis protein 1) and cIAP2 [109]. These E3 ligases create further large interaction surfaces by attaching K63-linked ubiquitin chains to components of the TNFR1 complex [110]. Unlike K48-linked ubiquitination, decoration of proteins with K63-linked ubiquitin chains does not lead to proteasomal protein degradation, but rather provides scaffolding functions by allowing the binding of additional signaling proteins. Another multifunctional E3 ligase is TRAF6, a receptor-proximal signaling component activated by IL-1 (interleukin-1) and TLR (toll-like receptor) ligands. Triggering of receptors for TNFα and IL-1α and of TLRs leads to activation of the NF-κB (nuclear factor kappa-light-chain-enhancer of activated B cells), JNK (c-Jun N terminal kinase [MAPK8]) and p38 MAPK signaling pathways [111,112]. Further downstream of TRAFs, the signaling cascades engage the adapter proteins TAB2/3 (TGFβ-activated kinase 1 [MAP3K7] binding protein 2/3), which bind to the protein kinase TAK1 (TGFβ-activated kinase 1), a central activator of TNFR, IL-1 and TLR-mediated signaling [113,114,115].

TAK1 (MAP3K7) is a mitogen-activated protein kinase kinase kinase (MAP3K), a class of enzymes that lead to the sequential and phosphorylation-dependent stimulation of several downstream kinases [113,116,117,118]. Another example for an important MAP3K is mitogen-activated protein kinase kinase kinase 3 (also called MAPK/ERK kinase kinase 3 or abbreviated MEKK3), which coordinates intracellular signals during inflammation and stress [119]. Up to six tiers in this cascade result in the activation of the **mitogen-activated protein kinase (MAPK) signaling pathway**. Mammals have four major MAPK signaling cascades, which are named according to their MAPK components: p38 α/β/γ/δ (p38), JNK1-3 (c-Jun N terminal kinase 1 to 3), ERK1/2 (extracellular signal-regulated kinase 1 and 2) and ERK5 [120]. The coordinated activation of these kinase networks controls all steps of pre- and post-transcriptional gene expression. In response to activation by the proinflammatory cytokine IL-1, the kinase TAK1 contributes to the inducible recruitment of the acetyl transferase CBP (CREB-binding protein) to the enhancers and promoters of inflammatory target genes mediating the subsequent acetylation of histone H3 at lysine 27 (K27), a mark for active enhancers [121]. In parallel, TAK1-mediated phosphorylation events lead to the activation of the **transcription factors** AP1 (activator protein 1) and NF-κB, two key mediators of the inflammatory gene response [122,123]. Of note, MAPK signaling cascades and regulatory ubiquitin ligases of the TRAF family also participate in the control of **post-transcriptional gene expression**. Particularly well described is the regulation of tristetraprolin (TTP; see below) and of KH-type splicing regulatory protein (KSRP), AU-rich element RNA-binding protein 1 (AUF1) and Hu-antigen R/ELAV-like protein 1 (HUR1) by p38, JNK or protein kinases downstream of both enzymes as schematically visualized in Figure 3 [103,111,124,125,126,127,128,129,130].

## 2. Selected Examples

As outlined above, mRNA biogenesis relies on series of nuclear and cytoplasmic processes. Intricate regulatory mechanisms control each step in the mRNA’s life cycle and allow for the integration of intrinsic and external signals. In the following, we will focus on selected examples that illustrate how post-transcriptional gene expression can be controlled by signaling cascades.

### 2.1. mRNP Remodeling Through Protein Modifications

#### 2.1.1. Roles of Post-Translational Modifications in mRNP Remodeling

PTMs are not only relevant in the signal transduction events leading the activation of transcription factors and the transcription process, e.g., by phosphorylation of the carboxy-terminal domain of RNA polymerase II [131], but also for the subsequent steps in mRNP function and composition [132,133]. Recent advances in mass spectrometry and analytical chemistry have allowed for the identification of >100 types of different PTMs decorating the side chains of amino acids. So far, only a limited number of different PTMs has been frequently detected in mRNP proteins, which may be due to several reasons: (I) many PTMs are not preserved in standard lysis buffers and can only be detected under specific conditions. (II) Most modifications occur in sub-stoichiometric amounts and can only be measured after enrichment of the modified peptides. (III) Detection of many PTMs requires focused experimental and bioinformatics analysis [134,135]. These reasons might contribute to the finding that for most mRNP proteins only few modifications are known including phosphorylation—the most abundant PTM—acetylation, SUMOylation, ubiquitination and methylation [132,133,136]. While many PTMs are introduced by writer proteins, interpreted by reader proteins and removed by erasers, other PTMs such as acetylation can also proceed by non-enzymatic mechanisms [137]. The dynamic life cycle of mRNPs requires their assembly, remodeling and dissociation. While some mRNPs form stable complexes that are amenable to structural analysis via crystallization or cryoelectron microscopy [138], other are highly dynamic and undergo a frequent exchange of protein or RNA components. This dynamic regulation involves modification of both principal mRNP components, namely the mRNA and the protein. At the level of RNA, nucleotide modifications and variations in RNA length give an important contribution for the sizing and composition of mRNPs [133,139].

In *S. cerevisiae*, several mRNP components are modified by PTMs that regulate mRNP dynamics. For example, ubiquitination of Yra1 by the E3 ligase Tom1 is necessary for mRNA export as this modification displaces Yra1 from mRNPs before their nuclear export [80,140]. Mex67 is recruited to the transcription side by interacting with Rsp5-ubiquitinated Hpr1, an interaction that protects Hpr1 from degradation [79]. Hpr1 is also sumoylated, and this PTM regulates the recruitment of the THO complex to mRNPs, which is important for the expression of acidic stress-induced genes [141]. Finally, the F-box protein Mdm30 ubiquitinates Sub2 targeting it for degradation, which enhances the recruitment of Yra1 to transcribed genes and thus promotes mRNA export [142]. Thus, several examples exist, in which PTMs on mRNP components regulate the assembly or dynamics of the mRNP.

In the remainder of this paragraph, we will focus on the dynamics and regulation of PTMs on the protein components of mammalian mRNPs. Due to space constraints we will only discuss the functional consequences of selected and representative PTMs on mRNP remodeling, which are also summarized in Table 1. A unifying characteristic of the affected RBPs is that they often assemble in membrane-less organelles (MLOs) by phase separation. The liquid-like nature of such mRNPs allows a dynamic exchange of components as well as fission and fusion events. MLO proteins often have large stretches of intrinsically disordered regions that frequently harbor low-sequence-complexity domains enriched in structure-breaking amino acids (Gly, Pro) or polar side chains (Arg, Gln, Glu, Ser, Lys) [143,144,145]. Extensive mutagenesis of proteins in the FUS (fused in sarcoma) family has revealed the relative contribution of individual amino acids for phase separation [145,146]. These studies demonstrated phase separation-promoting activities for tyrosine and arginine, whereas glutamine and serine residues were found to promote hardening. Modification of these amino acids can alter the charge or bulkiness of the side chains and thus affect the interaction with other proteins or RNAs.

Arginines are frequently found in their methylated form in mRNP proteins, with the modification occurring either as a monomethylation or as a symmetric or asymmetric dimethylation [160]. A restrictive function of arginine methylation for phase separation has been revealed in several studies. Expression of the arginine methylase PRMT1 (protein arginine methyltransferase 1) causes asymmetric dimethylation of arginines in the RNA helicase DDX4 (DEAD-box helicase 4), suppressing phase separation of this RBP [147]. Similarly, asymmetric dimethylation of arginines in the low complexity domain of hnRNPA2 reduces its phase separation by disrupting arginine-mediated contacts [148], and arginine methylation of the FUS protein negatively interferes with phase separation [149,150]. However, there are also examples showing a positive contribution of arginine methylation in phase separation, e.g., the symmetric dimethylation of the LSM4 (U6 snRNA-associated Sm-like protein 4) protein promotes P-body formation [152]. Beyond its contribution to phase transition, arginine methylation also controls the function of mRNPs. The CARM1 (coactivator associated arginine methyltransferase 1)-mediated arginine methylation of the mRNA stabilizing factor ELAVL1 (ELAV-like 1; also known as HUR1) contributes to the downregulation of SIRT1 (Sirtuin 1)-encoding mRNAs during the differentiation of human embryonic stem cells [153]. Arginine methylation also contributes to the co-transcriptional recruitment of the yeast pre-mRNA splicing factor Snp1 and its mammalian homolog U1-70K (SNRNP70) of the U1 snRNP to the spliceosome [154].

Collectively, many studies showed a functional relevance of arginine methylation for phase separation and RNP function, but these studies are experimentally challenging given that (I) the methylation-mimicking or -deficient status cannot be faithfully reproduced by amino acid mutations, (II) interference with writers or erasers of arginine methylation necessarily affects a large number of different substrates and thus does not allow conclusions on the function of one particular residue, and (III) antibodies specifically detecting individual arginine methylation sites are rarely available. Despite these caveats, PTMs on arginines are important regulators of phase separation. It will be very interesting to study in the future whether PTMs on other amino acids have similar regulatory effects.

Modification of lysines by the attachment of ubiquitin or SUMO (small ubiquitin-like modifier) results in a steric bulk that has a significant structural impact. In addition, many proteins contain interaction surfaces specifically recognizing ubiquitin or SUMO, thus allowing the formation of interaction meshworks [161,162]. SUMOylation occurs in many different mRNP components, thus affecting almost all aspects of mRNA metabolism including capping, mRNA processing, RNA editing and RNA binding by hnRNP proteins [136,163]. SUMOylation of the THO/TREX complex component Hpr1 specifically controls the association of this complex with mRNPs. Absent SUMOylation of Hrp1 results in improper mRNP assembly of a subset of acidic stress-induced transcripts that escape degradation [141]. Furthermore, SUMOylation interferes with the RNA editing enzymatic activity of ADAR1 (adenosine deaminase RNA-specific 1), which converts adenosine to inosine [156].

Changes in kinase-mediated phosphorylation patterns are very rapid and can occur within a few minutes, a feature making regulatory phosphorylation ideal for the regulation of fast processes. In addition, phosphorylation often serves as a primary PTM that controls subordinate modifications such as ubiquitination, as exemplified by sequential modification of IκB proteins in the NF-κB pathway [164,165]. The distinct regulatory nature of tyrosine and serine/threonine-based phosphorylation was revealed by ultra-deep phosphoproteome analysis [166]. Less than 1% of the identified phosphorylation sites are phospho-tyrosines and the number of phosphorylation events occurring on other proteinogenic amino acids are difficult to judge, as these modifications are relatively instable and methods allowing their preservation and detection have just recently been developed [134]. Phosphorylation necessarily increases the negative charge of the side chain and thus can affect steric and chemical properties of the modified residues and also regulates phase separation positively or negatively. An example of an inhibitory effect of phosphorylation on phase separation is seen for FUS [152]. The indirect control of FUS phase separation is mediated by its interaction with Hsp27 (heat-shock protein 27), as the stress-induced phosphorylation of this chaperone keeps FUS in the liquid phase [167]. Furthermore, the RNA-binding proteins FMRP and CAPRIN1 are found in MLOs, and their phosphorylation patterns control their phase separation propensity with RNA and thereby tune deadenylation and translation rates [157,158]. Similarly, the RNA-binding splicing factor TDP-43 (TARDBP) occurs in MLOs, and a phosphomimetic mutation of serine 48 to glutamic acid disrupts its polymeric assembly, discourages phase separation and disrupts its RNA splicing activity [159]. However, the signals and kinases regulating this phosphorylation still need to be identified.

Beyond the regulation of phase transition, phosphorylation also affects numerous functions of mRNP proteins. One example is provided by TTP, which binds to AU-rich elements (AREs), which are enriched in the 3’ UTRs of many cytokine-encoding mRNAs. TTP binding to its cognate mRNAs inhibits their translation and also leads to their destabilization, thus suppressing cytokine production (Figure 3). Stimulation of cells with lipopolysaccharide (LPS) results in the induction of p38-activated MK2/3 kinases (MAPKAP kinases 2/3), thus leading to TTP phosphorylation at serines 52 and 178, which in turn results in an inhibition of TTP-mediated mRNA destabilization [128]. LPS stimulation also triggers SUMOylation and subsequent nuclear translocation of the kinase IRAK2 (interleukin 1 receptor associated kinase 2). In the nucleus, IRAK2 phosphorylates SRSF1 to reduce binding to its target mRNAs and to promote nuclear export by ALYREF and NXF1-NXT1 [160].

These studies collectively show the important contribution of PTMs for various aspects of mRNP function. In the future, it will be important to determine the occurrence of signal-specific modifications for each protein using top-down proteomic approaches [168] and to clearly distinguish between constitutive and signal-regulated modifications. CRISPR-based methods will allow for testing the functional relevance of the modified amino acids at the level of the endogenous proteins. In addition, the identification of enzymes responsible for these modifications will be a prerequisite for the elucidation of specific signaling pathways impinging on mRNP function.

#### 2.1.2. Regulation of P-Body Factors by Post-Translational Modifications

P (Processing)-bodies are microscopically visible cytosolic mRNPs, which contain non-translating mRNAs and the major components of the 5′-to-3′ mRNA decay machinery. They are also largely devoid of translation initiation factors [169,170]. The number and size of P-bodies change when mRNA turnover is blocked at the steps of deadenylation, decapping or 5′ exonucleolysis [171]. P-body numbers also change dynamically in response to multiple external cues including osmotic stress, toxic chemicals, cytokine (IL-1) treatment or translational repression [172,173,174]. At present, the main function of P-bodies seems to be the (transient) storage of mRNA under unfavorable external conditions [139,170,175,176,177,178,179]. Constitutive P-body factors comprise CCR4-NOT, LSM1-7, the heterodimeric decapping complex with its subunits DCP1a (decapping mRNA 1a) and DCP2, various decapping activators such as EDC3, PAT1, DDX6 (also called Rck/p54, Dhh1p in yeast), EDC4 (also called Hedls, Ge-1, or RCD-8, absent in yeast), the 5′-to-3′ exoribonuclease XRN1 and 4E-T [180].

Despite numerous well-known modifications on RBPs (see Section 2.1.1), the number of publications describing direct modifications and their biological roles on proteins contained in P-bodies is still scarce. In this section, we will briefly review what is known about regulatory PTMs in central P-body proteins. It is important to distinguish observations in yeast from those in mammalian systems as the orthologous proteins can vary significantly. For example, the Dcp1a proteins of *S. cerevisiae* and *Schizosaccharomyces pombe* comprise 231 aa (26 kDa) and 127 aa (15 kDa), respectively, whereas the human and mouse variants are 582 aa (63 kDa) and 602 aa (65 kDa) in length. The large C-terminal extension of mammalian DCP1a has likely evolved to regulate additional functions. On SDS-PAGE, DCP1a usually migrates as a doublet as first observed in neuronal differentiation [181]. By systematic site-directed mutagenesis and by generation of phospho-specific antibodies, the upper form of DCP1a was found to contain DCP1a inducibly phosphorylated at serine 315 in response to interleukin-1, a prototypical proinflammatory cytokine [174]. Sustained JNK activation by the translational stressors anisomycin or sorbitol led to a dispersion of serine 315-phosphorylated DCP1a-positive P-bodies, which was also observed by ectopic expression of MEKK1 or TAK1, two MAP3Ks that strongly activate JNK (and p38 MAPK) [174]. Later on, it was shown that serine 315 was hyperphosphorylated during mitosis in parallel to the disassembly and (re)assembly of P-bodies [182]. The phosphorylation site around serine 315 is conserved in higher mammals and corresponds to a consensus MAPK phosphorylation motif (P-X-S/T-P; P, proline, S, serine, T, threonine, X, any aa). In line with this, serine 315 and the adjacent serine 319 can also be phosphorylated by the MEK1/2-ERK pathway during adipocyte differentiation or respiratory syncytial virus infection [183,184]. Besides its localization in P-bodies and the cytoplasm, DCP1a is also found to shuttle into the nucleus where it can act as a (comparably weak) transcriptional activator, or, by an unknown mechanism, as a suppressor of NF-κB transcriptional activity [174,185,186]. DCP1a is also modified by non-degradative lysine 63 (K63)-linked polyubiquitin chains, a modification that promotes the transient formation of large signaling hubs in multiple systems, including the NF-κB pathway [129,187]. K63-linked ubiquitin polymers are synthesized by the multifunctional E3 ubiquitin ligase TRAF6 (TNF receptor associated factor 6). The TRAF6 enzyme was shown to ubiquitinate DCP1a and to control DCP1a phosphorylation at Ser 315 [129]. Mutation of six C-terminal acceptor lysines of DCP1a affected P-body number and size, suggesting that these lysines alter the RNA and protein composition of P-bodies [129]. In all these studies, the functional roles of DCP1a modifications appear to be diverse and require the interactions with additional signaling and decapping proteins (e.g., TRAF6, JNK, DCP2). As a result, to date no clear-cut function of DCP1a has been identified that exclusively depends on a single regulated PTM.

In yeast, Dcp2, the catalytic subunit of the decapping complex, has been found to be phosphorylated at serine 137 by Ste20, a MAP4K upstream of several MAPK pathways that is distantly related to mammalian germinal center kinases (GCKs), which activate MAPK cascades in mammals [188,189,190]. Phospho-mimetic or inactivating mutations of serine 137 of Dcp2 stabilize or deregulate subsets of yeast mRNAs, suggesting that phosphorylation of decapping enzymes executes gene-specific control [188].

DCP2 not only tightly interacts with its regulatory subunit DCP1a, but also binds to EDC4 (enhancer of mRNA decapping 4) and XRN1 (5′-3′ exoribonuclease 1) to form a large multiprotein complex that couples decapping with 5′ exonucleolytic degradation of mRNAs (Figure 3) [191,192]. EDC4 was shown to protect the C-terminus of DCP2 from degradative proteasomal ubiquitination, a mechanism contributing to constitutive activity of DCP2 [193]. Somewhat similar, the E3 ubiquitin ligase malin (NHLRC1) is recruited to P-bodies to promote the (basal) degradation of DCP1a [194]. The knockdown of the deubiquitinating enzyme USP4 (ubiquitin specific peptidase 4) reduced the numbers of P-bodies, adding further evidence for ubiquitin chains as critical regulators of P-bodies [195].

The protein 4E-T (4E transporter; EIF4ENIF1) is required for the localization of the translation initiation factor eIF4E to P-bodies. 4E-T is phosphorylated by JNK at several proline residues and this promotes the formation of larger P-bodies upon oxidative stress [196].

The CCR4-NOT complex is phosphorylated by HIPKs (homeodomain interacting protein kinase), a feature that is shared with their yeast progenitor kinase Yak1 [197]. Little is known about phosphorylation or other modifications of EDC3, EDC4 and the LSM1-7 proteins.

It is well known that P-body number and size change dynamically in multiple conditions [169]. These observations are in contrast to the limited evidence available for signal-mediated regulation of P-body proteins by PTMs as summarized above. It is possible that the majority of regulatory steps in P-body assembly are provided by the sets of RNAs and their individual RBPs that localize to P-bodies. However, there is a large number of modifications found for the proteins mentioned in this paragraph from shotgun mass spectrometry proteomics experiments deposited in the PhosphoSitePlus database [147] whose functions have not yet been explored [198]. Comprehensive functional analyses of these sites will be crucial to solve the question if P-body factors largely assemble constitutively, or if their interactions, activities and organizations are controlled by direct modifications in response to external cues.

### 2.2. RNA Processing Changes in Response to External Signals

#### 2.2.1. Differential Regulation of microRNAs Upon Stress

miRNAs (microRNAs) are short single-stranded RNAs processed from longer RNA precursors with extensive secondary structure [199,200]. More than 1200 miRNA precursors have been identified in humans and mice to date [201]. miRNAs play a unique role in the cytoplasmic regulation of mRNA translation. As such, mature miRNAs are recruited to Ago (Argonaute) proteins to form the miRNA-induced silencing complex (miRISC). In this complex, miRNAs are the moiety used to scan for partially complementary mRNA sequences and affect their translation and/or stability in GW182-containing P-bodies [202] (Figure 4). Because perfect complementarity between a miRNA and its target mRNA is not required for regulation of translation, a single miRNA species is able to affect the production of several hundreds of target proteins at once [203]. By regulating mRNA translation, miRISC provides an essential level of regulation affecting many cellular regulatory pathways, including cell differentiation, proliferation and apoptosis. Indeed, altering the homeostatic levels of miRNAs is associated with many diseases [204].

miRNA precursors are predominantly transcribed by RNA polymerase II, and their expression can be dynamically regulated at the transcriptional level and exhibit tissue-specific patterns [201]. For example, in the context of immune activation by pathogens, several microRNAs can be rapidly induced by the nuclear translocation of NF-κB (e.g., miR-9, miR-21, miR-146, miR-155 [210]). In addition to transcriptional regulation, there is now growing evidence that miRNA biogenesis, relying on processing of the miRNA precursors by RBP complexes, can also be dynamically regulated. Primary miRNA transcripts (pri-miRNAs) are transcribed as long structured RNAs that are initially processed into ~70 nt long transcripts forming hairpin-like secondary structures (miRNA precursors or pre-miRNAs) by the nuclear microprocessor complex composed of the RNase III enzyme Drosha and the protein DGCR8 (DiGeorge syndrome critical region 8) [211,212] (Figure 4). The second step of miRNA maturation takes place in the cytoplasm and is conducted by the RNase III enzyme Dicer (encoded by *DICER1*), while being modulated by TRBP (TAR [HIV] RNA binding protein, encoded by *TARBP2*) and by PACT (protein activator of interferon-induced protein kinase EIF2AK2, encoded by *PRKRA*) in vertebrates [201]. Since pre-miRNAs are double-stranded, two different miRNAs can be generated from the same hairpin, referred to as 5p or 3p miRNA if they originate from the 5′ arm or 3′ arm of the pre-miRNA, respectively (Figure 4). The strand with the less stable 5’ end is usually loaded onto RISC and the other strand is discarded [201].

With the increased popularity of small RNA sequencing, it has become apparent that pre/pri-miRNA processing is quite diverse. As such, most mature miRNAs are not limited to a single length, but are rather exist as a mix of short RNA fragments varying between 18 to 28 bases (referred to isoforms of a miRNA or isomiRs) in vertebrates and in plants [201,213]—although we will focus on vertebrates in this section. In addition to miRNA end variations, specific internal miRNA variations can be added post-transcriptionally through ADAR (adenosine deaminase acting on RNA) editing [214], potentially affecting target interaction and the miRISC repertoire of targets—these are known as polymorphic isomiRs. Adaptor ligation, reverse transcription and amplification protocols can lead to biases in miRNA sequencing datasets and miRNA 5′/3′ end variations, as illustrated by studies relying on pooled synthetic miRNAs [215,216]. Nevertheless, there is ample evidence that beyond a technical artifact the prevalence of miRNA isoforms differs among cell types and tissues, where the miRNAs are expected to play different biological activities [217,218]. Studies comparing isomiRs indicate that isomiRs are more frequently seen in 5p than 3p miRNAs, suggesting a prevalent modulation at the step of Dicer processing [206] (Figure 4). This is in agreement with reports that while both 5′ and 3′ end length variations can be seen in miRNAs, 3′ end modifications are the most frequent [207,208]. Nonetheless, alternative processing at the level of Drosha can also lead to isomiR production [209,219]. Importantly, TRBP expression directly modulates isomiR cleavage by Dicer as recently reported in *Tarbp2*-deficient E15.5 mouse embryos [220].

Recently, it was discovered that miRNA termini can be dynamically regulated upon bacterial infection [208,221] and stimulation with IFN (type I interferon) [210,222,223]—which executes a potent antiviral effect in cells through the transcriptional modulation of hundreds of genes. We reported that selective abundant miRNA families displayed changes in isomiR stoichiometry such that longer isoforms were decreased by IFN, while shorter were increased [222]. Our data supported the involvement of the IFN-inducible 3′-5′ exonuclease PNPT1 (polyribonucleotide nucleotidyltransferase 1) in this effect, in line with a previous report [224]. Critically in this instance, it is the dynamic induction of PNPT1 expression upon IFN stimulation that regulates the rapid modulation of isomiR stoichiometry, thereby controlling their abundance and the regulation of their targets [222].

Independent reports confirmed our observations that isomiRs can be dynamically regulated [225,226,227], indicating that inducible isomiR processing is not limited to the context of immune responses to infection. A handful of examples support the concept that the length of miRNA ends can directly impact their function [207,228,229,230], but how this is broadly operating is poorly defined beyond the cases of 5′ end variations which impact target recognition (miRNA 5′ ends predominantly interact with their target mRNAs). Studies have demonstrated that 3′ end modifications can also directly impact miRNA function [207]. This concept is also reinforced by independent studies revealing 3′ end bases largely contribute to miRNA-target binding and regulation [231,232] and affect target repertoires [233]. Nonetheless, whether target regulation can be impacted upon a dynamic alteration of isomiR stoichiometry has not been clearly evidenced to date. Beyond the modulation of their target repertoire, changes in isomiR stoichiometry can impact miRNA function through the modulation of their intracellular levels, as seen with the case of miR-222-3p in the context of type-I IFN production [204].

The capacity to dynamically modulate miRNA processing, resulting in isomiR variations is therefore likely to result in different regulatory activities on mRNA targets, and strongly supports that key RBPs involved in miRNA processing can themselves be dynamically modulated in response to various stimuli. As previously mentioned, bacterial infections broadly impact isomiR levels in immune cells [208,221,222]. Bacterial infections rapidly modulate the expression of hundreds of genes [234], for instance through activation of NF-κB signaling upon bacterial recognition by innate immune sensors. Importantly, recent analyses of pre-miRNA processing indicate that >180 RBPs are involved in the modulation of specific pre-miRNA processing by Dicer [205], and it is likely that—similar to TRBP—these proteins are involved in the modulation of isomiR formation. We note that TRBP and several proteins shown to contribute to miRNA biogenesis (e.g., LGP2 (DHX58) [235]), or miRNA stability (e.g., the 5′-3′ exonuclease XRN1 [236]) were modulated more than two-fold upon *Salmonella* Typhimurium infection [234]. We speculate that dynamic regulation of RBPs controlling Dicer processing helps to fine-tune the stoichiometry of miRNA isoforms, thereby impacting their regulatory repertoire of targets.

Finally, miRNA end variations can also happen independently of pre-miRNA processing by Dicer or Drosha. As such, miRNAs can be modified by terminal nucleotidyltransferases such as TUT4 (terminal uridylyl transferase 4), TUT7 or TENT2 (terminal nucleotidyltransferase 2, also known as GLD-2), which elongate miRNAs by adding one or more 3′ terminal nucleotides (such as 3′ end uridylation or adenylation) [216,228,229,233]. Given that TUT7 expression is rapidly increased upon lipopolysaccharide stimulation of mouse macrophages [237,238], this would also be expected to directly contribute to the dynamic modulation of isomiRs’ 3′ ends.

Collectively, accumulating evidence supports the concept of a dynamic RBP-mediated modulation of miRNA biogenesis in the context of cellular stress such as pathogenic infection. This effect of RBPs impacts the stoichiometry of miRNA length variants, in turn modulating their activities on mRNA translation. However, further studies are needed to define the breadth of dynamic isomiR processing across cellular stresses, the signaling pathways that mediate these changes and the key RBP components underpinning these regulatory activities.

#### 2.2.2. Responding to Signaling via Alternative Splicing

Cellular processes are constantly monitored and regulated not only by the cellular status, but also through the external environment. Among the diverse molecular processes that are modulated by signaling, AS forms a critical aspect of gene regulation in plants and animals. Various external stimuli arising from the nutritional status, pathogen infection, chemical stimuli etc. can have an impact on AS [20,239]. The effect of the external environment is amplified in plants, which exhibit extensive phenotypic plasticity due to their sessile nature [240,241]. In this section, we will discuss examples of how external signals modulate plant growth and development through AS.

Plants flower at the appropriate time, taking into account both internal developmental and external environmental factors, to ensure maximum reproductive success [242]. Flowering time provides an excellent example, in which multiple facets of AS have been documented to play a role. Light and temperature are two key environmental factors that modulate flowering [243]. The model plant *Arabidopsis thaliana* is a facultative long-day plant, in which long days such as those seen in spring/summer promote flowering. Genetic analyses of flowering time identified multiple genes, including those that play a critical role in the response to environmental stimuli.

When exposed to warm temperature *A. thaliana* plants make flowers earlier compared to plants grown at lower temperatures [244]. Plants grown at 16 °C in short days (non-inductive light conditions) and shifted to either 25 °C in short days or 16 °C in long days (inductive light conditions) result in flowering plants. While both shifts result in floral induction, the signaling associated with the transition differed with light-mediated induction when shifted to long days and temperature-mediated induction when shifted to higher temperature. A microarray analysis revealed that genes associated with RNA processing (e.g., SR proteins) were differentially expressed specifically in response to temperature [244]. SR proteins are associated with splicing, which led to testing of flowering time-associated genes for AS [245,246].

Genetic analysis implicated a role for *FLM* (*FLOWERING LOCUS M*) in the temperature-mediated induction of flowering [244] (Figure 5). Interestingly, *FLM* is known to be alternatively spliced [247]. However, this AS event does not change the protein product, but rather modifies *FLM* expression through AS coupled with nonsense-mediated mRNA decay (AS-NMD) [248]. When plants are grown at 16 °C, *FLM* produces only a handful of transcript isoforms with the primary transcript being *FLM-β* [244]. The *FLM-β* transcript encodes functional FLM protein, which represses the key flowering time regulator *FT* (*FLOWERING LOCUS T*) suppressing floral transition. Ambient temperatures beyond 25 °C perturb splicing, resulting in the production of a multitude of splice variants at the expense of *FLM-β* [244,248,249]. The majority of these alternatively spliced transcript isoforms contain premature stop codons upstream of splice-junctions and long 3′ UTRs, which are signatures of NMD transcripts. Analysis of *FLM* splicing in *upf* (*up frameshift*) mutants that are defective in NMD revealed that *FLM* expression levels are regulated by NMD [248,250]. Elevated temperatures effectively lead to AS-NMD of the *FLM* transcripts, which in turn reduces the amount of functional FLM protein, resulting in a derepression of *FT* and hence earlier flowering [248,251]. This provides a nice example of how AS-NMD plays a significant role in conferring a phenotypic response at the organismal level to an external signal caused by environmental change.

CO (CONSTANS) is one of the primary proteins involved in floral transition and is essential to induce FT expression in long days. CO undergoes AS to produce protein products COα and COβ, which differ in their ability to bind to DNA [252]. The binding of COα-COα homodimers to the promoter of FT is stronger than that of COα-COβ heterodimers. In addition, the CO protein itself undergoes ubiquitin-mediated protein degradation involving interaction with the E3 ubiquitin ligase COP1 (CONSTITUTIVE PHOTOMORPHOGENIC 1). Degradation of the COα-COβ heterodimer by the COP1 proteasome is more efficient than for the COα-COα homodimer. Thus, AS of the *CO* transcript impairs the transcription of *FT* both by increasing the degradation of CO protein dimers and by reducing the binding to *FT* [252,253]. In essence, the effect of AS is mediated via diverse mechanisms both involving differential production of proteins as well as by regulating the levels of the functional mRNA being produced (Figure 5).

Beyond the immediate response to stimuli, splicing also provides an excellent opportunity to increase phenotypic diversification across diverse organisms [254]. Sequence variation can confer changes in splicing, which in turn can have strong phenotypic consequences of evolutionary significance from mild changes to complete loss-of-function phenotypes. Consequently, AS is among the key mechanisms that drive this phenotypic variation across accessions. *FRI (FRIGIDA)* and *FLC* are among the primary determinants of natural variation in flowering time in *A. thaliana* [255,256,257,258,259]. One of the mechanisms through which *FLC* expression is altered appears to be via mutations that modulate *FLC* splicing. A few strains of *Arabidopsis* (e.g., Bur-0, Ll-2) have mutations that modulate *FLC* splicing. The resulting mRNA is predicted to encode a truncated *FLC* protein that confers early flowering [255,259]. In addition, there are structural variations at the *FLC* locus including transposon insertions that modulate *FLC* expression levels [256,259,260].

AS can also buffer against mutations under certain conditions. Mutations in the *ICARUS1* gene lead to a temperature-dependent growth arrest at elevated temperatures, and several strains harbor loss-of-function alleles at this locus [261]. *ICARUS1* is alternatively spliced and the variant isoforms contain premature termination codons. However, in the case of certain deletions, which will typically lead to a loss-of-function, AS can compensate. Here, the frameshift induced through intron retention is restored by the deletion leading to a full-length protein [261]. While it is unclear how common such mechanisms are across genes or organisms, this example clearly demonstrates the power of alternative splicing to buffer against natural mutations that could lead to detrimental phenotypes.

## 3. Conclusions

After decades of research, the principles of signal transduction as well as post-transcriptional regulation of gene expression are well understood. Nevertheless, how the post-transcriptional steps of gene expression are controlled by signal transduction cascades in response to internal and external stimuli has remained largely enigmatic. This is exemplified by the nuclear export block of bulk mRNA under stress conditions, which first has been observed already 25 years ago [262]. However, neither the underlying molecular mechanisms nor the upstream signal transduction pathways of this stress response phenomenon are known to date. This will only be resolved by the joint interrogation of both the signal transduction events and the post-transcriptional gene expression changes they mediate. Only by bridging the two fields that are usually viewed separately will we be able to greatly enhance our understanding of how cells react to changing environmental including developmental conditions. This knowledge will contribute to our understanding of (patho)physiological states in humans in which dysregulated signal transduction events change gene expression output.

## Figures and Tables

**Figure 1 biomolecules-10-01310-f001:**
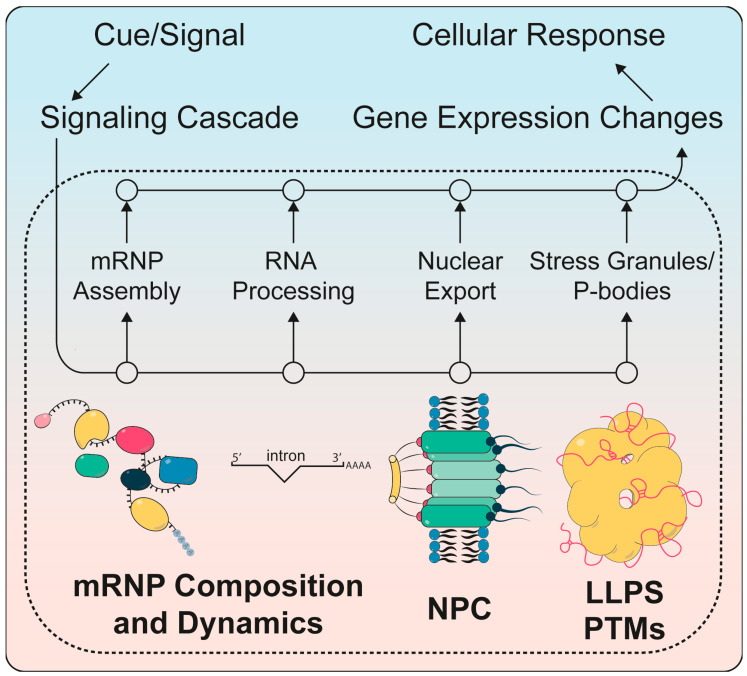
Regulation of post-transcriptional gene regulation by signaling cascades to generate the appropriate cellular response to an external cue. Extracellular cues or signals lead to intracellular signaling cascades (upper left) that regulate post-transcriptional steps of gene expression such as mRNP assembly, RNA processing, nuclear mRNA export through the nuclear pore complexes (NPCs) and the formation of stress granules and P-bodies (middle bottom). These changes are brought about by changes in RNP composition, post-translational modifications (PTMs) and liquid-liquid phase separation (LLPS). They reprogram gene expression in order to promote a cellular response (upper right) that is suitable for the initial cue.

**Figure 2 biomolecules-10-01310-f002:**
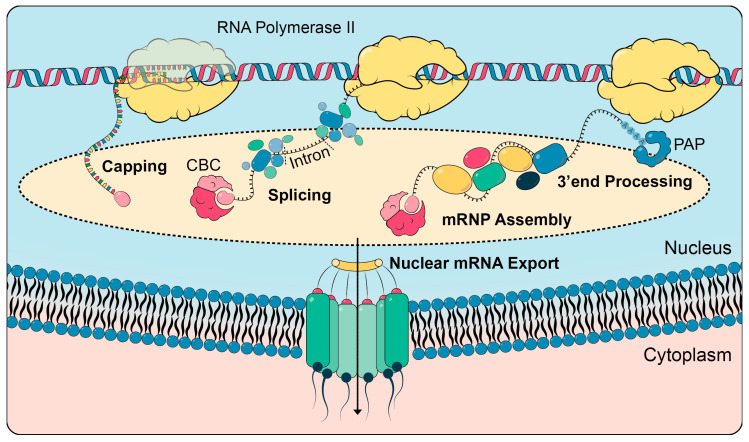
Nuclear mRNA processing, mRNP assembly and nuclear mRNA export. RNA polymerase II transcribes protein-coding genes and synthesizes the mRNA. The mRNA receives a cap at its 5′ end, to which the cap-binding complex (CBC) binds. Introns are spliced out by the spliceosome. At its 3′ end, the mRNA is cleaved and a poly(A) tail is added by poly(A) polymerase (PAP). Concomitantly, nuclear mRNA-binding proteins bind to the mRNA and package it into an mRNP. Only correctly processed and assembled mRNPs are exported from the nucleus to the cytoplasm.

**Figure 3 biomolecules-10-01310-f003:**
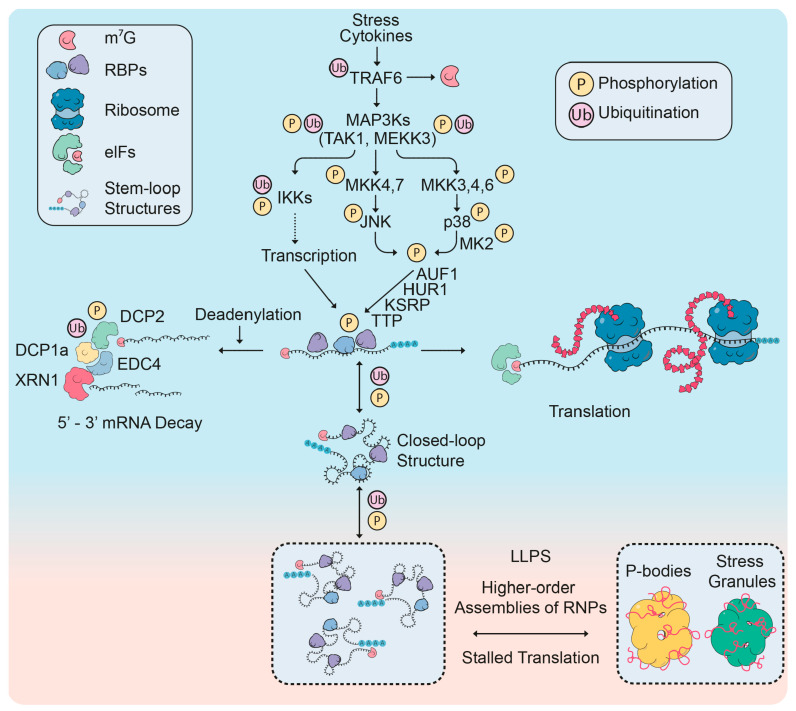
Coupling of signal transduction and (default) pathways affecting mRNP formation and mRNA fate. Stressful conditions and cytokines activate three canonical signaling pathways that primarily initiate mRNA transcription (through the IKK-NF-κB pathway) but simultaneously also promote mRNA decay and fast mRNA turnover (through the JNK and p38 MAPK pathways). The newly synthesized mRNAs are rapidly and efficiently unfolded and translated (right part) but can also be targeted for immediate decay by phosphorylation of prototypical RNA-binding proteins (RBPs) such as TTP, AUF1 and HUR1 (center). Decay proceeds stepwise through deadenylation, decapping and 5′ exonucleolysis (left part). The balance between mRNA translation and decay ensures fast and transient protein turnover in response to the above-mentioned triggers. Alternatively, mRNAs can transform into closed loop (circular) mRNPs or progressively fold into structures of higher complexity through liquid-liquid phase separation (LLPS) and formation of P-bodies or stress granules. These pathways allow mRNA storage and translational shut-off under unfavorable conditions. New studies now suggest that in addition to the (reversible) modifications of multiple RBPs, also the factors mediating decay, storage and higher-order assemblies of mRNPs are directly controlled by post-translational modifications, in particular phosphorylation by various protein kinases and non-degradative, K63-linked ubiquitin polymers by multifunctional E3 ligases such as TRAF6. For details and specific examples see text.

**Figure 4 biomolecules-10-01310-f004:**
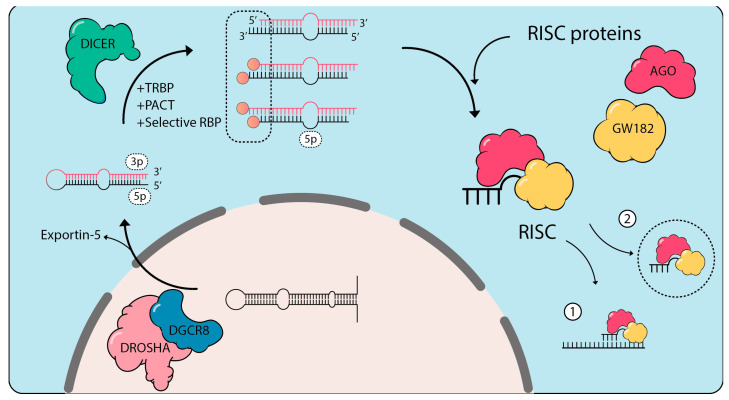
isomiR biogenesis. Primary microRNAs are predominantly transcribed by RNA polymerase II and are processed in the nucleus by the microprocessor complex formed of DROSHA and DGCR8. The resulting pre-miRNA hairpin short RNA is shuttled into the cytoplasm by Exportin 5 [201]. DICER and its binding partners TARBP and PACT subsequently process the stem loop of the pre-miRNAs with many RBPs selectively involved at this step depending on the pre-miRNA sequence [205]. Cleavage of the 5p arm by the resulting DICER complex will lead to 3′ end miRNA variations, which are the most frequent [206,207,208]. Nonetheless, 5′-end variations of the 3p arm can also occur. Similarly, variations of processing at the microprocessor level are also possible leading to 5′-end variations of the 5p arm and 3′-end variations of the 3p arm, respectively [209]. The resulting small RNAs are loaded onto AGO, and with GW182 control translational repression of partially complementary messenger RNAs (1). A proportion of miRISC complexes are also shuttled outside of the cells through exosomal export (2). The dynamic expression of RBPs modulates this process in response to stimulation.

**Figure 5 biomolecules-10-01310-f005:**
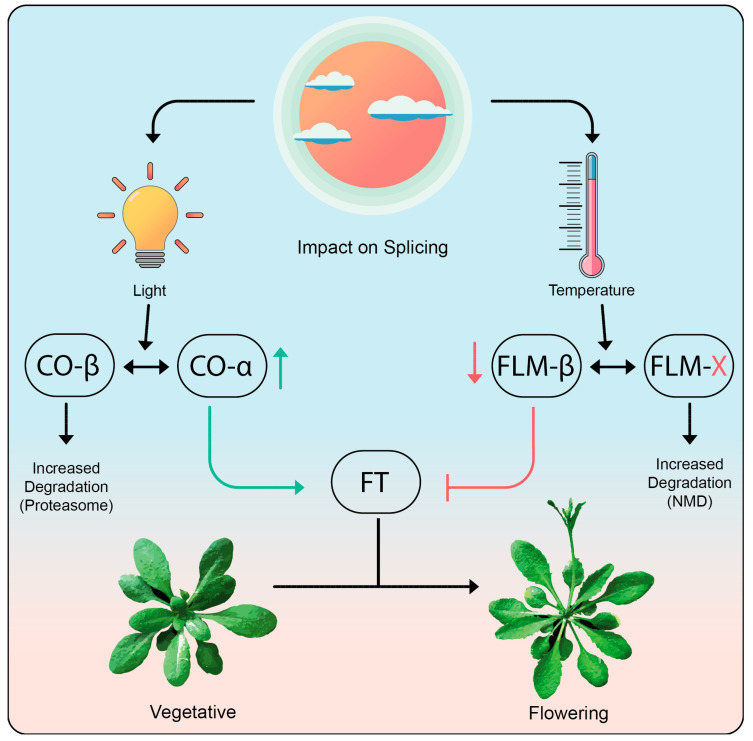
Role of alternative splicing in flowering time regulation in Arabidopsis thaliana. The transition from vegetative development to flowering requires the key gene FT, whose expression is induced by CO and repressed by FLM. Light-mediated changes in alternative splicing lead to an increase in CO-α, which promotes flowering, while CO-β-based dimers are degraded via COP1-mediated protein degradation. In contrast, higher temperatures promote alternative splicing of FLM leading to non-functional FLM variants (FLM-X), which are degraded through the NMD pathway. Splicing influence is shown in blue and inhibition is shown in red, while positive induction is shown in green.

**Table 1 biomolecules-10-01310-t001:** mRNP remodeling processes by post-translational modification of RBPs.

RNP	Modification	Functional Effect	Reference
DDX4	asymmetric dimethylation	suppression of phase separation	[147]
hnRNPA2	asymmetric dimethylation	reduction of phase separation	[148]
FUS	arginine methylation	reduction of phase separation	[149] [150]
	phosphorylation	reduction of phase separation	[151]
LSM4	symmetric dimethylation of	promotion of P-body formation	[152]
ELAVL1	arginine methylation	downregulation of SIRT1-encoding mRNAs	[153]
U1-70K (SNRNP70)	arginine methylation	recruitment to spliceosome	[154]
Hpr1	SUMOylation	promotes association with mRNPs	[141]
ADAR1	SUMOylation	impaired RNA editing activity	[155]
FMRP	phosphorylation	increased interaction with CAPRIN1 increased phase separation	[156] [157]
CAPRIN1	phosphorylation	increased interaction with FMRP increased phase separation	[156]
TDP-43 (TARDBP)	phosphorylation	decreased phase separation	[158]
TTP	phosphorylation	inhibition of TTP-mediated mRNA destabilization	[128]
SRSF1	phosphorylation	reduced binding to target mRNAs	[159]

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
