# Peer review of "Dynamic mRNP Remodeling in Response to Internal and External Stimuli"

_biomolecules, 2020, doi:10.3390/biom10091310_

Round 1

Reviewer 1 Report

The review article by Zarnack et al. addresses an important area of gene expression regulation, mRNP remodeling, which is much less often discussed as compared to events regulating transcription and protein function. While the review is generally well written with nice figures, I found it hard to follow due to the overall organization and the wide range of research areas covered (e.g. yeast, plant, and human systems discussing nuclear export, phase separation, miRNAs, signaling…) that often do not always focus on the exact topic of mRNP remodeling. In some ways, it reads like a lot of different text from different areas of expertise merged together without a unifying theme or effort to connect each section. I would recommend reorganizing and aligning the various sections with a clear set of questions in the introduction that are driving the field forward and link the areas discussed. Other comments:

  1. Abnormal to see a reference in an abstract, consider removing (see line 34).

  1. It is unclear from the layout of figure 1 and figure legend why the label “PTMs” is under the NPC? There is also no label in the figure or explanation for what the NPC is in the text for the reader. A non-expert would have no idea what this represents.

  1. The introduction should contain more broad/ general information on how internal and external stimuli are responsible for bringing changes in mRNP composition, localization, and stability in general. For example, there is a wealth of information in yeast and other systems on how changes in cell cycle, metabolic state, and environment alter gene expression post transcriptionally.

  1. The review would benefit from reorganizing the sections such that the basic gene expression pathway (sections 1.2 to 1.5) were written first followed by specific examples that address particular issues (e.g. signal transduction). This would also benefit the explanation of Figure 1 (see point #2).

  1. In the introduction, sub heading 1.1 principles of signal transduction, this section delves deep in to signaling cascades that activate inflammatory pathways, but why? This is a very specific example with no lead up as to the questions that are important to consider, which is hard to follow. There is not enough information (except figure 2) on how these signaling cascades bring about changes at the transcriptional or post transcriptional level such as at the level of mRNP remodeling.

  1. In figure 2, there is no poly A tail drawn on some of the mRNAs, is this on purpose? If so, why? Also, what is the difference between mRNAs with tick marks vs. mRNAs without tick marks? Generally, it would be beneficial if figures 1 and 2 were consistent with each other.

  1. In line 270 and 282, Nab2 is referred to as an SR-like protein. Nab2 has RGG motifs like Npl3, but no SR rich sequence, so is it an SR-like protein?

  1. In line 270, the text suggests that Pab1 is the dominant PABP in the nucleus, but Pab1 localization is mainly cytoplasmic, and literature suggest that Nab2 is the major nuclear PABP. As such, shouldn’t it be Nab2 that binds poly A tail in the nucleus?

  1. Test in line 296 is confusing, as Npl3, Hrb1, and Gbp2 are all SR proteins and export adaptors.

  1. In line 307-309, it is stated that there is little known about export under stress. However, there are various publications that address this, including reports of the Slt2 kinase acting on Nab2 during heat shock. Similarly, it is shown that phosphorylation of Nab2 reduces its interaction with Mex67 and alters its localization in the nucleus and thereby blocks the export of non-heat induced mRNAs. There are also reports of direct recruitment of Mex67 to mRNAs without export adaptors. All interesting areas of discussion that indicate there is some knowledge of what is happening during stress.

  1. In section 2.1.1, the main focus is on how post-translation modifications (PTMs) of RNA binding proteins (RBPs) alters the dynamicity and composition of membrane less organelles, but there is no information on how PTMs of RBPs change composition, structure, stability, and translation efficiency of mRNPs in general. For example, no information on the role of PTMs on RBPs such as Nab2, Yra1, and SR proteins etc. upon stimuli and how these PTMs promote mRNP remodeling. Additionally, this section is focused on membrane less organelles (MLOs) in metazoans, but the mRNA processing, export, and translation sections were mostly focused on yeast. This makes reading the review difficult with the changing gene names etc. Finally, given the focus on mRNA export and nuclear mRNPs, why is there is no mention of nuclear processing bodies?

  1. Given the title of the review, there is no discussion of how ubiquitination alters RNPs and MLOs, such as stress granules. Similarly, while there is mention of SUMO modification in lines 423-432, it is cursory and doesn’t provide details or mention how SUMOylation affects specific RNPs or MLOs.

  1. In section 2.2.1 there is no information on how altering stoichiometry of miRNA isomers ultimately affects their mRNA targets.

  1. In Section 2.2.2, RNA processing changes upon external stimuli could also include studies detailing changes in splicing upon nutrient starvation, stress etc. in yeast to connect the various sections.

  1. There is double space in line 479 after the word storage.

  1. In line 643, Typhimurium needs to be italicized and T needs to be lowercase.

  1. In line 659, RPB should be RBP.

Author Response

Reviewer 1

Comments and Suggestions for Authors

The review article by Zarnack et al. addresses an important area of gene expression regulation, mRNP remodeling, which is much less often discussed as compared to events regulating transcription and protein function. While the review is generally well written with nice figures, I found it hard to follow due to the overall organization and the wide range of research areas covered (e.g. yeast, plant, and human systems discussing nuclear export, phase separation, miRNAs, signaling…) that often do not always focus on the exact topic of mRNP remodeling. In some ways, it reads like a lot of different text from different areas of expertise merged together without a unifying theme or effort to connect each section. I would recommend reorganizing and aligning the various sections with a clear set of questions in the introduction that are driving the field forward and link the areas discussed.

We thank the reviewer for his/her interest in our work and the positive and constructive suggestions. We agree that the review covers a lot of ground and brings together a wide range of research areas. As seen in our point-to-point responses we have restructured parts of the review and altered the text according to the suggestions made.

Other comments: 

  1. Abnormal to see a reference in an abstract, consider removing (see line 34).
    As suggested we removed the reference to Figure 1 in the abstract (line 34).

  2. It is unclear from the layout of figure 1 and figure legend why the label “PTMs” is under the NPC? There is also no label in the figure or explanation for what the NPC is in the text for the reader. A non-expert would have no idea what this represents.
    As suggested we changed the layout of Figure 1: The nuclear pore complex is labeled with “NPC”, which is now explained in the figure legend. The label “PTMs” was moved below the stress granules / P-bodies.

  3. The introduction should contain more broad/ general information on how internal and external stimuli are responsible for bringing changes in mRNP composition, localization, and stability in general. For example, there is a wealth of information in yeast and other systems on how changes in cell cycle, metabolic state, and environment alter gene expression post transcriptionally.
    In response to this point and also the more general comments above, we now more clearly worked out how signal transduction, mRNP composition and RNA regulation come together throughout the review. In each section, we connect to examples from literature to illustrate what is already know in different systems.

  4. The review would benefit from reorganizing the sections such that the basic gene expression pathway (sections 1.2 to 1.5) were written first followed by specific examples that address particular issues (e.g. signal transduction). This would also benefit the explanation of Figure 1 (see point #2).
    We changed the order of the sections such that sections 1.2 to 1.5 are followed by section 1.1. This also led to a “swap” of Figures 2 and 3.

  5. In the introduction, sub heading 1.1 principles of signal transduction, this section delves deep in to signaling cascades that activate inflammatory pathways, but why? This is a very specific example with no lead up as to the questions that are important to consider, which is hard to follow. There is not enough information (except figure 2) on how these signaling cascades bring about changes at the transcriptional or post transcriptional level such as at the level of mRNP remodeling.
    We revised this section to point out that some general principles such as the control of signal thresholds, amplification processes, signal crosstalk and regulatory feed-back loops can be deduced from the well-studied pro-inflammatory signaling pathways (lines 305-307). Accordingly, we chose this signaling system as a general paradigm for rapid signal transduction processes that affect RNPs at all levels, rather than expanding on many different pathways. In order to provide more information on additional signaling regulators, we have now included MEKK3, KSRP, AUF1 and HUR1. Both text and Figure 3 (previous Figure 2) have been altered accordingly. Readers following the cited references will be guided to related fields of signaling that also participate in the regulation of mRNA synthesis or decay and mRNP formation.

  6. In figure 2, there is no poly A tail drawn on some of the mRNAs, is this on purpose? If so, why? Also, what is the difference between mRNAs with tick marks vs. mRNAs without tick marks? Generally, it would be beneficial if figures 1 and 2 were consistent with each other.
    We changed the design of figure 2 (now figure 3) so it is consistent with figure 1 as well as figure 3 (now figure 2).
    We changed the design of Figure 2 (now Figure 3) so it is consistent with Figure 1 and Figure 3 (now Figure 2).

  7. In line 270 and 282, Nab2 is referred to as an SR-like protein. Nab2 has RGG motifs like Npl3, but no SR rich sequence, so is it an SR-like protein?
    We agree with this reviewer and changed “nuclear SR-like protein” to “nuclear poly(A)-binding protein” (now line 209) and omitted “SR-rich” (now line 240).

  8. In line 270, the text suggests that Pab1 is the dominant PABP in the nucleus, but Pab1 localization is mainly cytoplasmic, and literature suggest that Nab2 is the major nuclear PABP. As such, shouldn’t it be Nab2 that binds poly A tail in the nucleus?
    We now mention explicitly that Pab1 is mainly localized to the cytoplasm and that Nab2 is thus the main poly(A)-binding protein in the nucleus (lines 233-237).

  9. Test in line 296 is confusing, as Npl3, Hrb1, and Gbp2 are all SR proteins and export adaptors.
    We made this passage more clear by expanding the one “over-loaded” sentence into two. It now reads: “Several mRNA export adaptors exist in yeast: the TREX complex components Hpr1, Yra1, Gbp2 and Hrb1, Npl3 as well as Nab2 [78-82]. In mammalian cells, SR proteins such as SRSF3 serve as mRNA export adaptors [31,83].” (lines 277-280).

  10. In line 307-309, it is stated that there is little known about export under stress. However, there are various publications that address this, including reports of the Slt2 kinase acting on Nab2 during heat shock. Similarly, it is shown that phosphorylation of Nab2 reduces its interaction with Mex67 and alters its localization in the nucleus and thereby blocks the export of non-heat induced mRNAs. There are also reports of direct recruitment of Mex67 to mRNAs without export adaptors. All interesting areas of discussion that indicate there is some knowledge of what is happening during stress.
    We already mentioned the direct recruitment of the mRNA export adaptor Mex67-Mtr2 to the mRNA during transcription, e. without export adaptors, in former line 307. We now expanded this half-sentence to make this point more clear (lines 295-297). In addition, we mention the regulation of mRNA export by Nab2 – or rather the kinase Slt2 – in stress conditions (lines 291-293).

  11. In section 2.1.1, the main focus is on how post-translation modifications (PTMs) of RNA binding proteins (RBPs) alters the dynamicity and composition of membrane less organelles, but there is no information on how PTMs of RBPs change composition, structure, stability, and translation efficiency of mRNPs in general. For example, no information on the role of PTMs on RBPs such as Nab2, Yra1, and SR proteins etc. upon stimuli and how these PTMs promote mRNP remodeling. Additionally, this section is focused on membrane less organelles (MLOs) in metazoans, but the mRNA processing, export, and translation sections were mostly focused on yeast. This makes reading the review difficult with the changing gene names etc. Finally, given the focus on mRNA export and nuclear mRNPs, why is there is no mention of nuclear processing bodies?
    As outlined above, we revised the manuscript to better connect the different parts. We also included further examples to illustrate principles regulating mRNP dynamics by PTMs in different organisms (g. lines 643-653). Due to space restrictions, we omitted an additional part on nuclear processing bodies.

  12. Given the title of the review, there is no discussion of how ubiquitination alters RNPs and MLOs, such as stress granules. Similarly, while there is mention of SUMO modification in lines 423-432, it is cursory and doesn’t provide details or mention how SUMOylation affects specific RNPs or MLOs.
    This point is well taken, however this review has already >100.000 characters excluding spaces (corresponding to 29 pages in its current format). We are afraid that the addition of a separate section on ubiquitination and further post-translational modifications would further inflate this review and thus make it too lengthy. We cover several aspects of the connection between ubiquitination and MLOs when we describe how TRAF6 and ubiquitination may regulate P-bodies.

  13. In section 2.2.1 there is no information on how altering stoichiometry of miRNA isomers ultimately affects their mRNA targets.
    We agree that this is an interesting aspect of isomiR function. How dynamic changes in isomiR impact the target repertoire is not defined yet in the literature beyond the reports that 5' end variations can affect targeting due to seed shift, and 3' end variations can also impact miRNA function (as stated in this section). We have amended the text to clarify this point (lines 905-909 and lines 910/911).

  14. In Section 2.2.2, RNA processing changes upon external stimuli could also include studies detailing changes in splicing upon nutrient starvation, stress etc. in yeast to connect the various sections.
    The second part of the review highlights selected examples. We therefore now cover the splicing changes upon nutrient starvation in yeast and related examples in the first part when introducing the different steps in RNA processing and their connection to signal transduction (lines 131-137 and lines 143-147). This also helped in our general effort to put more emphasis on the links between both aspects.

  15. There is double space in line 479 after the word storage.
    We deleted the superfluous space.

  16. In line 643, Typhimurium needs to be italicized and T needs to be lowercase.
    The term Typhimurium is not part of the species name, but specifies a serotype of Salmonella. Salmonella Typhimurium follows the official nomenclature by the Centers of Disease Control and Prevention (CDC) (see Brenner et al., 2000, J Clin Microbiol, 8: 2465–2467; doi: 10.1128/JCM.38.7.2465-2467.2000).

  17. In line 659, RPB should be RBP.
    We corrected this typo.

Reviewer 2 Report

In this review article, the authors summarized recent findings of the role of mRNP remodeling in response to internal and external stress. In particular, they focus on the current findings of the signaling pathways which modulate the de novo expression of mRNA and subsequent steps of posttranscriptional gene regulation. This review brings together a wide body of information that may be of use for researchers interested in this field. My specific comments for this manuscript are listed below.

Major comment:

Could the authors summarize columns about the mRNP remodeling through the post-translational modifications of RBPs mentioned in Section 2.1 as the table? It will help readers to understand their importance at a glance.

Minor comment:

On page 8, line 320: “The PIC scans along the 3’UTR for the start codon” should be “The PIC scans along the 5’UTR for the start codon”.

Author Response

Reviewer 2

Comments and Suggestions for Authors

In this review article, the authors summarized recent findings of the role of mRNP remodeling in response to internal and external stress. In particular, they focus on the current findings of the signaling pathways which modulate the de novo expression of mRNA and subsequent steps of posttranscriptional gene regulation. This review brings together a wide body of information that may be of use for researchers interested in this field. My specific comments for this manuscript are listed below.

Major comment:

Could the authors summarize columns about the mRNP remodeling through the post-translational modifications of RBPs mentioned in Section 2.1 as the table? It will help readers to understand their importance at a glance.

We thank the reviewer for this suggestion. We now summarize mRNP remodeling processes by post-translational modifications of RBPs as mentioned in section 2.1 in Table 1 (lines 717-727).

Minor comment:

On page 8, line 320: “The PIC scans along the 3’UTR for the start codon” should be “The PIC scans along the 5’UTR for the start codon”.

We corrected this typo.

Round 2

Reviewer 1 Report

The authors have addressed the points raised in my initial review.